# Health-Nutrients and Origin Awareness: Implications for Regional Wine Market-Segmentation Strategies Using a Latent Analysis

**DOI:** 10.3390/nu14071385

**Published:** 2022-03-26

**Authors:** Alessandro Petrontino, Michel Frem, Vincenzo Fucilli, Giovanni Tricarico, Francesco Bozzo

**Affiliations:** 1Department of Agro-Environmental and Territorial Sciences, University of Bari-Aldo Moro, Via Amendola 165/A, 70126 Bari, Italy; alessandro.petrontino@uniba.it (A.P.); vincenzo.fucilli@uniba.it (V.F.); francesco.bozzo@uniba.it (F.B.); 2Sinagri s.r.l., Spin off of the University of Bari-Aldo Moro, Via Amendola 165/A, 70126 Bari, Italy; 3Lebanese Agricultural Research Institute, Zone El Roumieh, Qleiat, Keserwan, Lebanon; 4Confcooperative Puglia, Viale Einaudi, 15, 70125 Bari, Italy; tricarico.g@confcooperative.it

**Keywords:** choice experiment, latent class model, lifestyles, wine

## Abstract

A healthy-nutrient wine has been recently developed by Apulian wineries (southern Italy), using autochthonous wine grapes cultivars, selected strains and specific processes of production. As such, this research elicits Italian wine consumers’ preferences towards this innovative Apulian wine with regard to additional labelling information associated with health-nutrients and the origin of grapes on the bottle of wine. For this purpose, a social survey based on the choice experiment approach is considered. The results reveal a heterogeneity of preferences among respondents for which the origin of wine grapes cultivars is the most appreciated (an average Willingness-to-Pay of EUR 6.57), thereby inducing an increase in their function utility, while the health-nutrients attribute is relatively less appreciated (an average Willingness-to-Pay of EUR 3.95). Furthermore, four class consumers’ cluster profile have been identified in respect to their: *(i)* behavior and propensity to wine consumption and purchase, *(ii)* health-claims importance on the wine bottle label, *(iii)* socio-economic characteristics and *(iv)* health conditions. This paper has marketing and public implications and contributes to an understanding of how additional information on the label of a wine bottle may affect the market-segmentation, influence wine consumers’ utility, protect their health and increase their level of awareness to wine ingredients labelling.

## 1. Introduction

In the European Union (EU), alcohol labelling for beverages is evolving. Its harmful consumption is considered the causal agent of many human diseases (cardiovascular 19%, liver cirrhosis 20%, cancer 29%) and injuries (19%) in the EU [1]. Consequently, a set of new actions has been proposed by the European Commission (EC) on the label of alcohol beverages in accordance with the United Nations Sustainable Development Goals and with its Europe’s Beating Cancer Plan. Among these actions, more importance has been placed on alcohol labelling declarations, mainly linked to nutritional facts and health warnings. It is mandatory to adopt the declaration of such information by the end of 2020 and 2023, respectively [2]. This nutritional declaration, except for the alcohol content, is actually facultative on European alcohol beverages such as wine [3]. The latter and health issues have always aroused conflicting opinions. On the one hand, the consumption of wine induces enthusiasm and health benefits, but on the other hand and owing to its alcohol degree and excessive consumption, wine induces heath concerns (i.e., liver or pregnancy damage), can lead to violence and causes mortality and morbidity [1]. Despite these conflicting opinions, its effects on the cardiovascular system and the antioxidant action of the substances contained such as polyphenols are evident [4].

However, the European Food Safety Authority [5] has expressed a negative opinion about the use of health claims on the label. Referring to the EFSA opinion n. 9(4), 2082 of 2011 claims associated with polyphenols from red wine or derived from it were not authorized. In particular, the reason for non-authorization was: “*Non-compliance with the Regulation because on the basis of the scientific evidence assessed, this food is not sufficiently characterized for a scientific assessment of this claimed effect and the claim could not therefore be substantiated*”. Nevertheless, a more recent work [6] attempted to summarize the current findings about the positive influence of wine consumption on human organ function, chronic diseases, and the reduction in damage to the cardiovascular system.

Italian wine labels usually comprise the following information: winery/producer, brand name, vintage vineyards, region, country, bottle size, alcohol volume, and abbreviations such as: VdT (*Vino da Tavola*/Table wine); IGT (*Vino a Indicazione Geografica*/Geographical Indication; DOC (*Vino a Denominazione di Origine Controllata*/Controlled Designed of Origin); DOCG (*Vino a Denominazione di Origine Controllata e Garantita*/Controlled and Guaranteed Designation of Origin. By the end of 2022, nutrients declaration, based on average values of producers, will become mandatory for all wine produced and sold with the European Union market. In Italy, wine labelling in terms of nutrients awareness, health information, alcohol content warning, and certification of origin, are voluntary, unregulated and heterogeneous [7]. Hence, several studies highlight that careful labelling or health warning [8], effective labelling with mixed text and image claims [9], framing messages and a low level of visibility of some warnings claims [7,10], represent important tools to fill the information gap between consumers and producers and, to enhance the rational consumption of wine.

Indeed, other studies promote the idea that the nutritional information in labels is viewed as one of the most effective ways to promote healthier food or beverages such as wine choices among consumers [11,12,13,14]. The nutritional properties and health benefits of wine [15,16], “naturalness” features [3], sustainability in terms of biodiversity in vineyards [17], environmental aspects of wine production processes [18,19], cultural, geographical territory, country of origin, area of production of wine grapes cultivars [20,21], traditional landscape [22], proximity production [23], have received the increasing interest of researchers.

Moreover, current wine consumption in Italy involves more and more young people and women. The trend has moved from a habitual consumption of medium-low quality wine to a more sporadic consumption of higher quality wine that is determined by the origin and low alcohol content [24]. Furthermore, using a best-worst technique [25], “testing the wine previously” and “matching food” are, in rank order, key determinants of Italian choice of wine, followed by “origin (country and region) of the wine”, “personnel knowledge about the wine”, “information on back label”, “grape variety” and, “recommended by someone else”. However, “medal/award, information on the shelf, an attractive front label, alcohol level (below 13%) and promotional display in store” are ranked as the least (worst) effective determinants that influence Italian wine preferences. Additionally, “previous trial and recommended by someone else” constitute the key determinants of worldwide consumer choice of wine, followed by “brand, grape variety, food matching and medal/awards”. “Low alcohol (below 13%) and “promotional strategies” also constitute the least weighted determinants [25].

Tracing back to the ancient period, the Italian climate has been recognized as a suitable environment to cultivate grapes. From 2020–2021, the average national production area of wine grapes has been estimated to be 646,985 ha. For the same period, Italy produced 7.72 million tons of wine grapes per year, inducing an average yield of around 12 Tons/ha. This production is distributed among grapes for wines with protected designation of origin (43%), grapes for wines with protected geographical indication (24%) and grapes for other wines (33%). Furthermore, the average national production of wine amounts to 5030 million liters [24], with a consumption volume of 2395 million liters [26] and an export value of 2913 million euros in 2019, for which Western European countries were the main destination of Italian wine [27]. Therefore, each Italian region produces its typical wine derived from local cultivars. As can be assumed, the Veneto region produced the highest volume of wine (23.20%), followed by Apulia (19.00%) and Emilia Romagna (15.35%) in 2019. As such, it is clear that the Apulia region, which is taken as the case study in this work, is particularly suited for grapevine cultivation and boasts some local cultivars (*Primitivo*, *Negroamaro*, *Bombino nero*, *Susumaniello* and *Notardomenico*), yielding wines with a high content of antioxidants. In 2021, Apulia accounted for 13.54% and 21.2% of the Italian wine-grapes area and share of production, respectively [24].

In this regard, within the project “*Domina Apuliae*”, funded by the Apulia Region (Appendix B), an innovative mode of wine production was assessed, with the collaboration of four Apulian wine cooperatives. The technical experimentations were based on the use of the cultivars mentioned above, selected yeast strains and specific processes capable of preserving antioxidant power, low alcohol content and high nutritional and health value. In this context, this study was carried out to assess the Italian wine consumers’ preferences towards this innovative mode of production via the consideration of two attributes: extrinsic (regional territory/Apulia red wine grapes cultivars) and intrinsic (heath and nutrition claim) information, graphically represented on the labels of the final products. Specifically, this study addressed three interconnected research questions: (i) what are the Apulian wine consumers’ profiles? (ii) what are their preferences towards these attributes? and (iii) what is their Willingness-to-Pay (WTP) for the presence of such additional information on the label of the bottles of wine?

To the best of our knowledge, no previous Italian studies have investigated wine preferences heterogeneity by investigating (i) combined regional wine grapes cultivars and graphically represented health attributes and (ii) the lifestyle and health status of consumers, as potential market-segmentation tools in wine choice. As such, this study has practical implications and seeks to support Apulian wineries in their appropriate and efficient promotional and marketing diversification strategies implementation as well as the premium price of the final wine products. Additionally, this study has public health implications in terms of supporting and urging the mandatory implementation of nutrients and health warning labels on wine bottles so as to affect drinking behavior and raising social awareness. To this purpose, an econometric Choice Experiment (CE) technique has been used by means of Multinomial Logit (MNL) and Latent Class Models (LCM). As such, the other added-value purpose of the present paper is to enrich the scientific literature on the choice modelling of wine consumers’ perception. In the following sections, we describe how the CE has been implemented in this study.

## 2. Materials and Methods

CE has been widely used in food marketing and in particular to investigate wine consumers’ choice preferences, develop cluster profiles and, measure WTP [3,7,16,17,28,29]. In line with these studies, CE was implemented here in five steps as follows: (i) selection of attributes and assignment of levels, (ii) choice of an experimental design, (iii) construction of choice set, (iv) social-choice survey and (v) measurement of consumers’ preferences using an MNL, the development of cluster profiles using LCM and the calculation of WTP in each model.

### 2.1. Selection of Attributes and Assignment of Levels

To define the attributes and their corresponding levels, a focus group approach was adopted in this study based on the needs of Apulian wineries in order to implement effective promotional and marketing strategies. As such, three attributes were retained: *(i) price* (with 4 levels of changes: EUR 2.50, EUR 7.50, EUR 10.00, EUR 15.00), *(ii) Local* grape vines cultivars (with two levels: wine made or not with Apulia vinegrapes: *Negroamaro, Primitivo, Bombino nero, Uva di Troia, Notardomenico or Susumaniello*) and, *(iii) Health* (with two levels: wine with absence or presence of high antioxidant properties due to the presence of a high content of polyphenols, specific process of production and selected yeast strains capable of preserving them). For the monetary attribute that the citizens are willing to pay as shown in Table 1, the Italian wine prices ranged from EUR 2 to 10 per bottle of 0.75 L in 2019 [27]. A level of EUR 15 per bottle of 0.75 L was agreed on by the focus group to be considered in this research in order to include averages values across a wider range. This attribute was considered as a discrete variable in the CE experiment. The last two attributes, considered as dummy variables, were also illustrated by corresponding symbols (Table 2) that aimed to help the respondents in their decision-making process [30,31]. This visual symbol [32] allowed the respondent to contemplate the attribute(s) under consideration [33] and supported the verisimilitude of the options [34]. In the adopted approach (CE), wine consumers may straightforwardly select a choice set.

### 2.2. Choice of an Experimental Design and Construction of Choice Set

Regarding the experimental design, the combination of the above attributes and their corresponding levels provided 16 possible scenarios (2^2^ × 4). According to a D-efficient Bayesian design, which ensures a maximization of statistical efficiency by minimizing the D-error [35,36], the 16 reasonable alternatives were divided into two blocks including eight choices per set (Table 2). Each choice set, graphically designed [7], consisted of three columns reflecting the two alternatives (Option A, Option B) plus an opt-out alternative (option C). It is assumed that the respondent has to select the option that leads to a maximization of the total utility of wine consumption under a budget constraint.

### 2.3. Social-Choice Survey and Data Collection

Italian consumers’ preferences were explored through an online social-choice questionnaire that was disseminated using social media and posters with a QR code directing a reader to an online questionnaire link, which was placed in busy places. The collection of data continued for a period of 4 months (from June to September 2019), involving 439 (Appendix C) valid wine respondents (as general public) in different cities of the Apulian Region. The online survey [28] targeted both sexes and all ages ranges. The questionnaire had an introduction in which the theme is presented and explained to the respondent and a further four sections. The first section explored behavioral questions with regard to respondent’s consumption and purchase propensity of wine. The second section included a simulation of the purchase of wine in which the block of eight set of choices were presented as described above. The third section revealed their preferences towards the origin, health and nutrition information on the label of the wine bottles. The fourth section aimed to collect data on their socio-demographic and economic characteristics as well as on their health status.

### 2.4. Measurement of Consumers’ Preferences

#### 2.4.1. Econometric Data Analysis

Each attribute included in our study constitutes one component of the multi-functionality (utility) of wine consumption. The latter can thus be expressed as follows:U_nj_ = V_nj_ + *ɛ*_nj_(1)
where:

n are the users (respondents); j are the alternatives (choice set); V_nj_ is a function of observable attributes of wine consumption, known as the deterministic component of U; *ɛ*_nj_ is a function of the non-observable features of the wine and respondent-level variation in tastes, unknown, considered as stochastic part of U and treated as random error.

V_nj_ can be expressed by:V_nj_ = *β’*x_nj_ = α + *β_1_*x_1n_ + *β_2_*x_2n_ + … + *β_m_*x_mni_ + *ɛ*_nj_(2)
where:

x_nj_ represents the attributes of the alternatives; *Β’*, *β*, represent the coefficients of the attributes of the alternatives. They reveal the weight of preference for each attribute level, as well as trade-offs in monetary value.

As mentioned above, the respondent is assumed to select the alternative which is associated with the highest utility (benefit or satisfaction). Thus, the probability that n-th respondent chooses the i-th alternative from a choice set is:P_ni_ = Prob (U_ni_ > U_nj_) ∀_j_ ≠ i= Prob (V_ni_ + *ɛ*ni > V_nj_ + *ɛ*_nj_) ∀_j_ ≠ i= Prob (*ɛ*_nj_ − *ɛ*_ni_ < V_ni_ − V_nj_) ∀_j_ ≠ i(3)

In our study, we can assume that respondents are heterogeneous in their preferences and that consumer behaviors depend on additional factors beyond those that are directly observable (individual characteristics). LCM captures preference heterogeneity across classes but assumes homogeneous parameter estimates within each class [37]. It is based on the assumption that the studied population is divided into different unobserved/latent classes concerning the attributes and levels [38]. Furthermore, LCM investigates whether consumers’ preferences differ according to some socio-economic characteristics and opinions for health and nutritional information that would be used on the label of wine bottles.

As such, LCM can help to segment the wine market and to estimate model parameters together with the selection of the more appropriate socio-economic and physical characteristics (covariates) of respondents able to influence their behavior, therefore the decision-making process. The identified segments (Q *q* classes) highlight different consumer preferences and the sensitivity to the attributes of the proposed product in connection with socio-demographic, attitudinal and physical characteristics. A step-by-step comparison process was used to select the most fitting covariates and the most appropriate number of classes in order to improve the model statistical properties. Concerning the respondents’ cluster profiles, the LCM was used with a four class-decision to ensure the best representation of the target wine consumers’ market. As such, a model fit statistics information criteria (IC: maximum log likelihood, minimum Bayesian Information Criteria/BIC and minimum Corrected Akaike Criteria/CAIC) was used for this purpose [3,39]. The changes in values of IC are reported in Table 3. By gradually adding the number of classes, the model presents the most optimum fit improvement in terms of stability, sensitivity and specificity [29,40]. In formal terms, the conditional choice probability of finding the consumer *i* in the class *q* for the observed alternative *j* is:(4)$$\pi_{ij|q}=\frac{\exp\left(\beta_{q}^{\prime }x_{ij}\right)}{\sum_{q=1}^{Q}\exp(\beta_{q}^{\prime }x_{ij})}$$
where *x_i_* denotes a set of characteristics that are associated with class membership and *β_q_* are specific class-related coefficients to estimate [41]. The conditional probability that consumers i chooses the alternative *j* is:(5)$$\pi_{ij}=\sum_{q=1}^{Q}\pi_{iq}\pi_{ij|q}$$

Finally, in order to best explain the choices of consumers, the estimation of the parameter values is carried out through the maximization of the log likelihood function:(6)$$lnL=\sum_{i=1}^{N}ln\left[\sum_{q=1}^{Q}\pi_{iq}{\left(\prod_{t=1_{i}}^{T_{i}}\pi_{it|q}\right)}^{y_{ij}}\right]$$
where *y_ij_* is one or zero if the consumer *i* chooses the alternative *j* or not, respectively.

#### 2.4.2. Willingness-to-Pay (WTP)

WTP assesses a premium price compared to a conventional wine. It reflects the amount that respondents are willing to pay for each proposed intervention. It is the ratio between the estimated parameters for selected attributes and the negative of the parameter estimated for the selling price: each ratio reflects the average contribution a consumer would pay for local grape cultivars and for health claims to be used on the label of wine bottle. As such, the calculation of WTP is calculated as follows:(7)$$WTPa=-\frac{\beta a}{\beta p}$$
where *WTP_a_* is the willingness to pay for the attribute a, while βa and βp are the estimated coefficients related to each attribute, respectively.

## 3. Results

### 3.1. Consumers’ Cluster Profiles: Market-Segmentation

The consumers’ profile (Appendix A has been categorized into four classes based on *(i)* behavior and propensity to wine consumption and purchase, *(ii)* health claims importance on the wine bottle label, *(iii)* socio-economic characteristics and *(iv)* health conditions of respondents as follows:

*Class 1: female, single or divorced, wine drinkers* (42% of total respondents). This is the second highest of the classes for which the majority of its members consume more than once a week (66.48%) and more than half-liter each day (38.9%), with an average of around one glass of wine per week consumed usually outside of meals. Consumption of wine usually takes place at their property, parents or friends’ home (50.27%—highest class across all). Wine is mostly bought from non-specialized stores such as hypermarkets, supermarkets, hard discount stored or via the Internet (64.87%—highest class). The quality of wine as a choice factor importance is most appreciated by them, followed in succession by the following factors; origin of wine grapes, type of cultivars, price, label content and alcohol degree. With regard to the level of importance placed on health and nutrition claims on the label, the latter obtained the highest importance for grapes coming from Apulia autochthones vines. On the contrary, a label with low alcohol degree is least appreciated across all classes. With respect to their socio-economic characteristics, this class is mainly dominated by female (59%), single, divorced or separated (82.17%) with an average age of 45.5 years. Their level of education is around 17 years, making them an educated class among the others. Their body mass index is normal. Their diseases are generally related to the cardiovascular system (4.86%—highest among any class). Their general level of health is 1.08, reflecting very good health.

*Class 2: socio-ecological oriented, relatively highly educated and slightly wine consumers* (15% of total respondents). Among any class, this group presents the lowest wine consumption frequency (58.46% more than once a week) and consumption metrics (73.85% less than half-liter each day), with an average of one glass of wine per week, usually consumed outside of meals. Not only do they form the class of the highest usual wine drinkers at restaurants, pubs or wine bars, but also the second highest class in terms of purchasing wine from specialized shops (43.07%). The quality of wine is the most important to them. Price and label content of wine bottles have a similar importance to them. Among all classes, they highly appreciate wine produced with a low content of sulfites, with high content of antioxidants and, via an integrated production system with low ecological impact. With respect to the socio-economic characteristics, this class present a few different distributions between male and female, with an average age of around 42 years. This class also presents the highest level of education (17.18 years) among all segments. With regard to heath conditions, a very low number of respondents (1.53%) present diseases related to the cardiovascular system, while this class reveals the highest case of diseases related to the respiratory system, explaining their tendency to select healthy and ecological wine production. As such, this class is ranked lowest in terms of the general health status level (1.29).

*Class 3: wine consumers, regional cultivars oriented, relatively less educated* (18% of total respondents). Among the classes, this segment consumes the most wine (67.94% more than once a week, 41.02% more than half-liter each day, 1.74 glass/week usually outside of meals and outside their proper, parents or friends’ homes). Members of this class mostly purchase wine from non-specialists wine shops (61.54%). Their preferences level for the price and label aspects of wine bottles are more or less similar to class 2. In addition, wine produced with ancient Apulian vines is most important to them. Mainly single or divorced (65.39%), they are relatively female-dominant with an average age of 41 years and a relatively low level of education (15.85 years) compared to all segments. However, they constitute the largest class for which their annual income ranges less than EUR 20.000 per year. Concerning their health status, they present the lowest health status level (1.52) and the highest class suffering from immune system diseases.

*Class 4: specialist drinkers, quality label content oriented and relatively oldest* (25% of total respondents). As usual drinkers, wine consumption here occurs slightly equally between their proper homes and outside (62.16% more than once a week, 1.24 glass per week, usually out of meals). They purchase wine from specialists wine shops at a higher rate than any other class. Furthermore, they attribute the highest level of importance to quality, alcohol degree, and label content. Additionally, they hold the least importance for all information related to health claims importance on the wine bottle label, except for information regarding wine produced with a low alcohol degree. This class is male-dominant (54.95%), the oldest (47 years), single, divorced or separate (75.68%) with the lowest level of education (15.55 years) of any class. More than its half members (53.15%) have an annual income between EUR 20.100 and 35.000. Their body mass index (24.48) falls within the normal or healthy weight range while 6.30% of them suffer from metabolism disease.

### 3.2. Consumers’ Preferences for Attributes: Multinomial Logit and Latent Class Models

The MNL reveals that all attributes significantly affect wine consumers’ preferences, implying that the presence of symbols (Table 2) presents higher utility than their absence or no symbols on the label of the bottles of wine. Among all attributes, the origin of grapes symbol (wine made from Apulia’s grapes) was the most appreciated by Apulian respondents, inducing an increase in their functional utility, while they appreciated the health and nutrition symbol to a lesser degree. Their coefficients have positive and significant values, suggesting that these features have a high conditional impact on choice probability. As expected, the price attribute for using this kind of information is highly significant and negatively correlated with consumers’ preferences, indicating a higher disutility function for labelling on the wine bottle as shown in Table 4, in which class 4 was considered as the reference class. The LCM analysis reveals the probabilities of classes, with values of 42%, 15%, 17% and 25% for classes 1, 2, 3 and 4, respectively. Furthermore, all classes present a negative and significant price coefficient. The latter for “Local” is positive among all classes and significant for class 1, highly significant for classes 2 and 3, while it is not significant for class 4, suggesting that for 25% of the statistical sample of respondents, “local” appears not to be a determinant factor for their wine purchase and consumption. Concerning “health”, this attribute is estimated as significantly appreciated by classes 2 and 4, while for classes 1 and 3, it is comparatively inappreciable and significant. Overall, the positive and significant Asymmetric Specific Constant (No buy) reveals that members of any class, except for class 4 (negative and significant coefficient), are highly disposed to changes from the current situation in terms of additional information on the label of a wine bottle. Members in classes 2 and 3 show significant but negative coefficients estimates for age, indicating that older respondents of these classes are not affected by the presence of additional information on the wine bottle, while members with a high level of education in classes 1 and 2 show a high preference towards the introduction of the additional information. Additionally, the factor quality, as a purchase choice parameter of respondents, is negative and insignificant among all classes. Therefore, the perception of polyphenols in red wine helping to prevent the oxidation caused by free radicals is positive and insignificant, except for in class 2.

### 3.3. Consumers’ Willigness-to-Pay

As Figure 1 shows, the WTP of wine consumers for additional information on the label of wine bottle tends to be heterogeneous across classes. In class 4, consumers are willing to pay the highest premium price (EUR 10.43) for labelling regarding the regional origin of wine, followed by class 1 (EUR 7.59), class 3 (EUR 5.29) and class 2 (EUR 2.98). With respect to health and nutritional information, all members’ classes are willing to pay a premium price range between EUR 2.46 and 10.26, except for class 1 (representing the highest size of the sample), for which its members tend to not appreciate this kind of labelling feature. This can be explained by the fact that they do not have enough awareness or knowledge about the beneficial aspect of rational wine consumption.

## 4. Discussion

### 4.1. Health-Nutrients

With respect to the presence of the health-nutrients symbol, our results reveal a relatively lower significant perception in the purchase and consumption of wine. Concretely, findings from the LCM confirm that not all respondents attached a positive utility to the health symbol on the bottle of wine. These results might not largely support previous studies, such as Mueller et al. [28] who found a high acceptance of the ingredients list on the wine label, Pabst et al. [40] who also prompted the idea that wine ingredients induce positive consumption utility, and Barreiro-Hurle et al. [42] who provided an insight into the functional properties of wine that highly affect the repurchase intent as well as Costanigro et al. [43] who found that differentiating wine labels (i.e., low sulfites level) increases wine consumption. Furthermore, our study highlights that the level of perception with regard to the alcohol content of wine is differentially valued among all respondents. Previous papers also assumed the importance of the labelling of wine with a low alcohol content. For Bucher et al. [44], wines with a reduced alcohol content can be an interesting product for a variety of stakeholders and may offer benefits for consumers while having the potential to reduce alcohol consumption and therefore contributing to a reduction in alcohol-related harm. Later, the same author [44] deepened the study of how people evaluate low-alcohol wine and if the reduction in alcohol and the information that a wine is low in alcohol influences consumption. The mean consumed amount across the conditions compared in the study (normal wine and low alcohol wine) did not differ but participants were willing to pay more for the normal wine compared to the low-alcohol wine. Saliba et al. [45] provide useful information for wine producers and marketers in terms of people in the population who are interested in low-alcohol wine by describing the results of an Australian survey. The results showed that those most likely to purchase low-alcohol wine were female and reasons for preferring a low-alcohol wine included lessening the adverse effects of alcohol. Furthermore, our results suggest that consumers of class 1 (female-dominated, single or divorce, wine drinkers), representing the highest size of the sample, are subjects of interest for local wineries and decision makers in order to enhance their health perception and consciousness [3] through health awareness promotions strategies and public communication plan.

### 4.2. Origin of Cultivars Grapes

With regard to the origin of cultivars grapes, our analysis highlights a highly significant preference of respondents for the Apulian regional vine grapes cultivars used to produce local red wine. These results are in line with previous studies [46,47,48,49] for which the region of origin or the origin of wine (domestic versus imported) was considered an important attribute for product market-segmentation, and wine drinkers attraction. As such, providing such information increase purchase intent and induces a relatively high utility function for wine drinkers in the study area. As such, our results are in line with previous studies in a similar southern European country (Spain) in which, it has been highlighted, wine consumption is affected by the designation of origin [50] and regional wine is more appreciated than national wines [28] for Spanish wine consumers. The “well-known region” significantly increases the selection of wines [46,51] and, the “quality-territory” [52] concept expresses a high market strategy for local wineries. Furthermore, the indications of wine grapes cultivars [53], local wine production, area and appellation of origin [54], induce positive utility and WTP.

Here, we assumed that consumers might link the importance of grapes vines cultivars to the attractive wine landscape or to the evocative landscape [55] that induces a high ability for the selection of wine. Moreover, the results expressed with regard to WTP highlight the opportunity for the wine sector in the Apulia region to generate economic income on the basis of the adoption of typical Apulian vine grape cultivars and via the promotion of additional nutritional quality information on the bottles of wine, particularly the common product, (VdT) which present an average of around 65% of the total Apulian wine production for 2016–2019.

Although consumers of class 2 are the least likely to pay a premium price for the regional origin information, they show an interest in the sustainable production of vine grapes. Our findings suggest that this kind of consumer, representing the lowest size of respondents, would be more interested in labelling related to organic, sustainability and environmentally friendly vine and wine production. However, despite the fact that a low ecological impact system of production provides fewer yields than the intensive production of vine grapes, our results are in harmony with many previous studies [28,56,57,58,59,60,61] which found that “eco-friendly and sustainable claims” labelling is more attractive for wine consumers and increases their perception. As such, our study suggest supporting a promotion strategy related to the landscape and sustainability features [61,62] and an eco-friendly manner of production on wine labels [56].

### 4.3. Segments Characterisitcs

Overall, our results support a previous study on wine market segmentation. Rodrigues-Donate et al. [8] also found the heterogeneity of socio-demographics in wine consumers’ decision. Additionally, Bruwer et al. [28] also assigned five wine consumers segments on the basis of their wine consumption behaviour and lifestyles. The same authors demonstrated the importance of market-segmentation and highlighted the fact that segment characteristics provide relevant data for an effective wine differentiation strategy. Additionally, Zhllima et al. [49], using a CLM, showed that a five-class model best fits wine consumers. The same authors highlighted that this model constitutes an effective for use by marketers to define their market-segment strategy. Janssen et al. [63] highlighted the preference heterogeneity segments of respondents regarding the price and origin of wine.

### 4.4. Strengths of the Study, Its Limitations and Directions for Future Research

Concretely, this study demonstrates the potential effectiveness and the necessity of various forms of effective and careful communication and the important of ingredient [62] quality factors [64] as well as geographic information [65,66] on wine labels public policy and for the strategy of wineries. Our results might help to fill the gap in the literature about consumers’ interests in buying wine with a certain linkage of health properties and local variety used in the process. In relation to consumers, in addition to conventional socioeconomic characteristics, their physical condition, particularly associated with age and gender would probably be considered in addressing a market approach. The geographical scope of this research might limit the extrapolation of the observed results across the country. Thus, future research should use CE on wine on a national geographical scope with more detailed levels of nutrition, ingredients and vine grapes cultivars. The elicitation of wine tourists’ preferences could also be covered by future research owing to their potential positive effect on the consumption and purchase of Italian wine.

## 5. Conclusions

The mandatory labelling of alcoholic beverages such as wine ingredients is currently evolving in the European Union [7]. In this context, this study elicits Italian consumers wine preferences heterogeneity regarding the health-nutrients and regional origin of vine grapes cultivars on the wine bottle. These attributes are driven by different respondents’ attitudes. As such, the findings, explored above, provide practical implications in terms of the market-segmentation of wine with regard to the behavior and propensity to wine consumption, the importance of purchase factors, the preferences of respondents for additional labelling, their demographic and socio-economic characteristics and their health condition in the Apulia region. From a value point of view, this research is the first to integrate the latter variables into the domain of wine market segmentation using LCM. The health condition status of sample respondents considered as part of the life style segmentation constitutes added-value for cluster profiling and product differentiation and provides more effective information than demographic and socio-economic variables. Furthermore, the topic explored here is relatively new in the wine market and may be relevant for vineries interested in differentiating their products.

## Figures and Tables

**Figure 1 nutrients-14-01385-f001:**
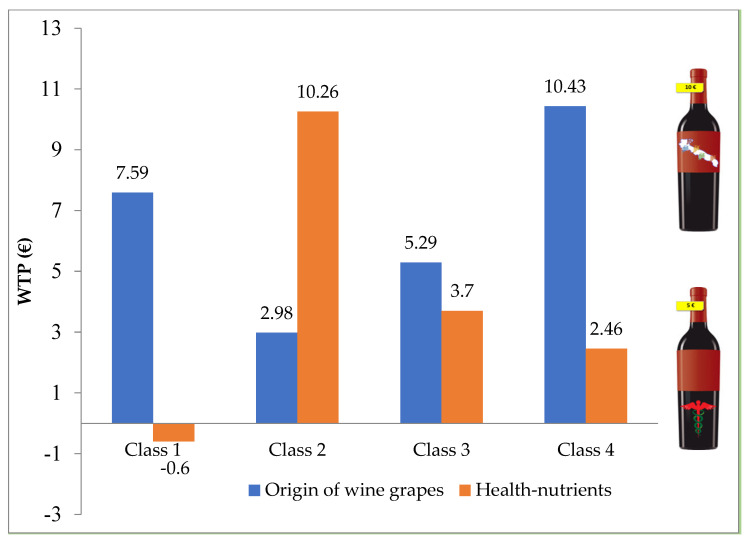
WTP for the introduction of effective communication symbols on red wine label.

**Table 1 nutrients-14-01385-t001:** Attributes and levels used to elicit Italian wine consumers’ preferences.

	Attributes		
	Price in EUR (Bottle of 750 mL)	Intrinsic Information for the Origin of Grapes (Apulia Region)	Intrinsic Information for Health & Nutrition
**Levels of attributes**	2.5	Not present on the bottle	Not present on the bottle
	7.5		
	10	Present on the bottle	Present on the bottle
	15	(Wine made with Apulian vine grapes: *Negroamaro, Primitivo, Bombino nero, Uva di Troia, Notardomenico or Susumaniello*)	(Wine with high antioxidant properties due to the presence of high content of polyphenols, specific process and selected yeast strains capable of preserving them)

**Table 2 nutrients-14-01385-t002:** Example of a set choice.

Attribute	Option A	Option B	Option C
Origin of grapes Code: Local	Present on the bottle	Not present on the bottle	Neither A nor B I will not pay a premium price for additional information/labelling strategies (No buy)
Health-nutrients Code: Health	Not Present on the bottle	Present on the bottle	
Price (EUR)	10 EUR/bottle 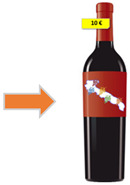	10 EUR/bottle 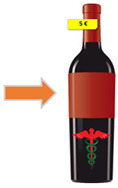	
Which option do you prefer?	□	□	□

**Table 3 nutrients-14-01385-t003:** The information criteria values for models with 1 to 6 classes.

	Multinomial Logit	2-Class	3-Class	4-Class	5-Class	6-Class
Log likelihood	−3345.69	−3085.01	−2978.48	−2854.1	−2815.03	−2788.48
CAIC	6699.40	6198.00	6005.00	5776.10	5718.10	5685.00
BIC	3362.02	3142.16	3076.45	2992.84	2994.63	3008.90
R^2^Adj	0.121	0.199	0.225	0.257	0.266	0.272
Average classes probabilities	100%	66%	66%	42%	41%	40%
		34%	15%	15%	17%	14%
			18%	17%	23%	17%
				25%	8%	4%
					11%	19%
						6%

**Table 4 nutrients-14-01385-t004:** Multinomial Logit Model and Latent Class Model estimates.

KERRYPNX	MNL		4-LCM							
			Class 1		Class 2		Class 3		Class 4	
	Coef.	[z-Value]	Coef.	[z-Value]	Coef.	[z-Value]	Coef.	[z-Value]	Coef.	[z-Value]
Class probability	-		42		15		17		25	
Price	−0.17 ***	−22.41	−0.37 ***	−14.48	−0.24 ***	−7.32	−0.09 ***	−3.62	−0.08 ***	−3.73
Local	1.17 ***	21.20	2.81 *	16.54	0.69 ***	2.79	0.47 ***	3.51	0.94	4.58
Health	0.14 ***	2.62	−0.27 ***	−1.94	2.66 ***	8.63	−0.46 ***	−3.63	0.24 ***	1.36
No buy	0.49 ***	6.21	1.25 ***	4.86	0.98 ***	2.93	2.73 ***	8.12	−1.54 ***	−6.13
Co-variates										
Constant	-		−2.95 **	−2.04	−4.26 **	−2.32	0.56	0.38	-	-
Age	-		−0.01	−0.81	−0.04 **	−2.20	−0.50 ***	−3.18	-	-
Education ^1^	-		0.17 ***	3.19	0.22 ***	2.64	0.02	0.50	-	-
Autochtone ^2^	-		0.92 ***	3.65	0.34	1.11	0.57 *	1.89	-	-
Oxidation ^3^	-		0.04	0.20	0.96 ***	3.45	0.20	0.89	-	-
Model statistics										
LL Function	−3346		−2854							
Pseudo-R^2^	0.12		0.26							
Observations	3512		3512							
Respondents	439		439							

Note: ***, **, * ==> Significance at 1%, 5%, 10% level, respectively. The final model configuration derives from a gradual approach in which the iterated inclusion of covariates is accompanied by a verification of the improvement of the goodness of the model. ^1^ “Education” refers to the education level in terms of the number of years; ^2^ “Autochthone” refers to grapes that come from Apulian autochthonous vines; ^3^ “Oxidation” refers to its level of importance (1: low; 2: medium; 3: high) given by respondents as additional information to be used on the label of the bottled red wine. In fact, polyphenols of red wine help to prevent the oxidation caused by free radicals.

## Data Availability

Not applicable.

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
