# Peer review of "Health-Nutrients and Origin Awareness: Implications for Regional Wine Market-Segmentation Strategies Using a Latent Analysis"

_nutrients, 2022, doi:10.3390/nu14071385_

Round 1
Reviewer 1 Report
This study uses the latent class method to analyze the consumers' willingness to pay for labeling of origin and health information for wines from Apulian. It is suggested that the authors clarify the method used to define latent classes.
- The notation of equation 4 is not clearly defined. Please show the prior and posterior probability of latent class model.
- It is important for readers to understand the rationale and method used for identifying the latent classes. Please show the method used to identify the classes in this research.
- In Line 390, a typo
- Since latent class model is selected, please show the insight of this model in more detail.
- How to find in the real world consumers belong to the latent classes defined in your research? How to apply your empirical results to the consumers in the real world?
Author Response
Dear Reviewer,
Please refer to the upload file.
Thank you
All the best

Reviewer 2 Report
Dear Authors,
I find the topic of the article submitted to Nutrients journal original and interesting. I also appreciate the concept of the study, however, the way of presenting the problematic of the paper raises some doubts and reservations, which I will present below.
Abstract and Keywords
The list of keywords is too long - the most important should be selected.
Introduction
This chapter is somewhat chaotic. The Authors discuss wine labelling, the importance of nutritional information and research projects carried out in Apula (this part of the introduction is too extensive) and the importance of their own study. There is a definite lack of information on the broad determinants of consumer choice of wine (in Italy and other countries).
There is also a lack of a clearly stated purpose of the study and research hypotheses (questions or problems). I think that rethinking these issues would help to organize the content presented in the Introduction. Presenting the discussed issues in a broader context would also provide more universal significance of the obtained results.
Materials and methods
The first subsection of this part of the article seems superfluous, because it does not contribute much to the methodology.
Description of the concept of the research is very extensive and I had some problem with understanding the precise idea and the course of the study. Perhaps it is a matter of presentation - it should be considered to organize the information presented (distinguishing the various stages). Perhaps it should considered creating a diagram presenting the scheme of the research. I think that these are "technical" issues.
The description mentions focus group research but there is no information about participants of the research. The same applies to the survey - I think information about the respondents should be included in the Appendix. Additionally, it is unclear how the sample was selected - was it a quota selection?
It is also necessary to address the issue of ethical standards. In the survey respondents are asked about their health status, this is sensitive data, so I do not understand the information given by the Authors at the end of the text: Institutional Review Board Statement: Not applicable and Informed Consent Statement: Not applicable.
Results
It seems that it would be more reasonable to first characterize the identified segments and then describe the diversity of their preferences. It would also be advisable to present the sociodemographic characteristics of the segments in a table.
Discussion
The first (rather extensive) paragraph of the discussion is devoted to describing the significance of the study conducted, which is hardly the topic of this chapter. The discussion is then given to health issues, the aspect of origin, and labeling. The discussion omits the issue of segment characteristics, without relating the results obtained to previously conducted studies.
Conclusions
This chapter is far too broad, and the information on the results is only a small part of what it contains and is additionally very general. The rest of this chapter is once again a presentation of the significance of the study, as well as its limitations.
I suggest that the chapter be more specific - what has been established as a result of the study, what has resulted from it, and (possibly) what specific applied significance the results have. I don't understand the references (62-65) cited in this chapter in relation to one's own results.
Information on the strengths of the study, its limitations, and directions for future research should be provided in a separate chapter.
In my opinion, the strength of the article is its problematic and interesting methodological approach. Other aspects need further development.
Author Response
Dear Reviewer,
Please refer to the attached file.
Thank you
All the best
